# Ethnic differences in early onset multimorbidity and associations with health service use, long-term prescribing, years of life lost, and mortality: A cross-sectional study using clustering in the UK Clinical Practice Research Datalink

**Fabiola Eto**[1]*, **Miriam Samuel**[1], **Rafael Henkin**[2], **Meera Mahesh**[3],
**Tahania Ahmad**[1], **Alisha Angdembe**[2], **R. Hamish McAllister-Williams**[4,5,6],
**Paolo Missier**[7], **Nick J. Reynolds**[7], **Michael R. Barnes**[2], **Sally Hull**[1], **Sarah Finer**[1☯],
**Rohini Mathur**[1☯]

1 Wolfson Institute of Population Health, Queen Mary University of London, London, United Kingdom,
2 William Harvey Research Institute, Queen Mary University of London, London, United Kingdom, 3 Barts
and The London School of Medicine and Dentistry, Queen Mary University of London, London, United
Kingdom, 4 Translational and Clinical Research Institute, Newcastle University, Newcastle, United Kingdom,
5 Northern Centre for Mood Disorders, Newcastle University, Newcastle, United Kingdom, 6 Cumbria,
Northumberland, Tyne and Wear NHS Foundation Trust, Newcastle, United Kingdom, 7 Newcastle
University, Newcastle, United Kingdom

☯ These authors contributed equally to this work.

* f.eto@qmul.ac.uk

**Citation:** Eto F, Samuel M, Henkin R, Mahesh M,
Ahmad T, Angdembe A, et al. (2023) Ethnic
differences in early onset multimorbidity and
associations with health service use, long-term
prescribing, years of life lost, and mortality: A
cross-sectional study using clustering in the UK
Clinical Practice Research Datalink. PLoS Med
20(10): e1004300. https://doi.org/10.1371/journal.
pmed.1004300

School, UNITED STATES

## Abstract

### Background

The population prevalence of multimorbidity (the existence of at least 2 or more long-term
conditions [LTCs] in an individual) is increasing among young adults, particularly in minority
ethnic groups and individuals living in socioeconomically deprived areas. In this study, we
applied a data-driven approach to identify clusters of individuals who had an early onset mul-
timorbidity in an ethnically and socioeconomically diverse population. We identified associa-
tions between clusters and a range of health outcomes.

### Methods and findings

Using linked primary and secondary care data from the Clinical Practice Research Datalink
GOLD (CPRD GOLD), we conducted a cross-sectional study of 837,869 individuals with
early onset multimorbidity (aged between 16 and 39 years old when the second LTC was
recorded) registered with an English general practice between 2010 and 2020. The study
population included 777,906 people of White ethnicity (93%), 33,915 people of South Asian
ethnicity (4%), and 26,048 people of Black African/Caribbean ethnicity (3%). A total of 204
LTCs were considered. Latent class analysis stratified by ethnicity identified 4 clusters of
multimorbidity in White groups and 3 clusters in South Asian and Black groups. We found

**Data Availability Statement:** The Clinical Practice Research Datalink (CPRD) does not allow the sharing of patient-level data. The structure and format of the CPRD data set is available at: https://cprd. com/sites/default/files/CPRD%20GOLD% 20Full %20Data%20Specification%20v2.0_0.pdf. The data that support the findings of this study are available from CPRD and access is subject to approval from an Independent Scientific Advisory Committee (ISAC). The data were used under license for the current study. The list of long-term conditions and respective code lists used in this study are available at: https://github.com/Fabiola-Eto/MULTIPLY-Initiative. The poLCAParallel R package is available at: https://github.com/QMUL/poLCAParallel/releases/tag/v1.1.0. This research utilised Queen Mary's Apocrita HPC facility, supported by QMUL Research-IT. http://doi.org/10.5281/zenodo.438045.

**Funding:** This work (and salary costs to SF, RM, SH, FE) was supported by MRC (MR/S027297/1). "Multimorbidity clusters, trajectories and genetic risk in British south Asians, 2020-2023". Link: https://gtr.ukri.org/projects?ref=MR%2FS027297%2F1. RM is supported by Barts Charity (MGU0504). Additional support for this and related work (SF, MRB, NJR, PM, RHM, RH, AA, MS, FE) is from NIHR 31672 AI-MULTIPLY, 2022-2025, and NIHR 202635 (SF, NJR, MRB, PM). The funders had no role in study design, data collection and analysis, decision to publish, or preparation of the manuscript.

**Competing interests:** FE receive salary for this work by MRC (MR/S027297/1). SF and RM receive salary contributions for their work on the Genes & Health programme, by a Life Sciences Consortium that includes Astra Zeneca PLC, Bristol-Myers Squibb Company, GlaxoSmithKline Research and Development Limited, Maze Therapeutics Inc, Merck Sharp & Dohme LLC, Novo Nordisk A/S, Pfizer Inc, Takeda Development Centre Americas Inc. RM is supported by Barts Charity (MGU0504). RM has received consulting fees from AMGEN unrelated to this work. RHM has received fees from American Center for Psychiatry & Neurology United Arab Emirates, British Association for Psychopharmacology, European College of Neuropsychopharmacology, International Society for Affective Disorders, Janssen, LivaNova, Lundbeck, My Tomorrows, OCM Comunicaziona s. n.c., Pfizer, Qatar International Mental Health Conference, Sunovion, Syntropharma, UK Medical Research Council and Wiley; grant support from National Institute for Health Research Efficacy and Mechanism Evaluation Panel and Health Technology Assessment Panel; and non-financial

that early onset multimorbidity was more common among South Asian (59%, 33,915) and Black (56% 26,048) groups compared to the White population (42%, 777,906). Latent class analysis revealed physical and mental health conditions that were common across all ethnic groups (i.e., hypertension, depression, and painful conditions). However, each ethnic group also presented exclusive LTCs and different sociodemographic profiles: In White groups, the cluster with the highest rates/odds of the outcomes was predominantly male (54%, 44,150) and more socioeconomically deprived than the cluster with the lowest rates/odds of the outcomes. On the other hand, South Asian and Black groups were more socioeconomically deprived than White groups, with a consistent deprivation gradient across all multimorbidity clusters. At the end of the study, 4% (34,922) of the White early onset multimorbidity population had died compared to 2% of the South Asian and Black early onset multimorbidity populations (535 and 570, respectively); however, the latter groups died younger and lost more years of life. The 3 ethnic groups each displayed a cluster of individuals with increased rates of primary care consultations, hospitalisations, long-term prescribing, and odds of mortality. Study limitations include the exclusion of individuals with missing ethnicity information, the age of diagnosis not reflecting the actual age of onset, and the exclusion of people from Mixed, Chinese, and other ethnic groups due to insufficient power to investigate associations between multimorbidity and health-related outcomes in these groups.

## Conclusions

These findings emphasise the need to identify, prevent, and manage multimorbidity early in the life course. Our work provides additional insights into the excess burden of early onset multimorbidity in those from socioeconomically deprived and diverse groups who are disproportionately and more severely affected by multimorbidity and highlights the need to ensure healthcare improvements are equitable.

## Author summary

### Why was this study done?

- Most studies of multimorbidity focus on older adults, and only a few have investigated multimorbidity in younger populations.
- The prevalence of multimorbidity is increasing among young adults, particularly in minority ethnic groups and individuals living in socioeconomically deprived areas.
- There is evidence showing that individuals with socioeconomic vulnerability experience poorer health outcomes, such as lower quality healthcare provision, premature death, and higher mortality rates.
- The association between early onset multimorbidity and poor health outcomes in ethnically and socially diverse populations in England remains underinvestigated.

support from COMPASS Pathways and Magstim. PM is funded by the National Institute for Health Research (NIHR) Newcastle Biomedical Research Centre, by NIHR AIM AI-MULTIPLY, and by NIHR ADMISSION (MR/V033654/1). NJR is funded by the National Institute for Health Research (NIHR) Newcastle Biomedical Research Centre, by the NIHR Newcastle In Vitro Diagnostics Co-operative and NIHR AIM AI-MULTIPLY. N.J.R. is also a NIHR Senior Investigator, (Senior Investigator Award) NIHR200168. NJR reports grants from PSORT industrial partners as listed (http://www.psort.org.uk/); other research grants from GSK Stiefel and Novartis. MRB is funded by the National Institute for Health Research (NIHR) AIM AI-MULTIPLY Consortium (NIHR203982). MRB reports research grants from Benevolent AI, Janssen and Novartis. TA is funded by the NIHR Applied Research Collaboration North Thames Award (NIHR 200163). AA is funded by the NIHR (31672 AI-MULTIPLY, 2022-2025). RH is funded by the Health Data Research UK (grant ref: LOND1).

**Abbreviations:** BNF, British National Formulary; CHD, coronary heart disease; CKD, chronic kidney disease; COPD, chronic obstructive pulmonary disease; CPRD, Clinical Practice Research Datalink; HES-APC, Hospital Episode Statistics Admitted-Patient Care; IMD, Index of Multiple Deprivation; LTC, long-term condition; YLL, years of life lost.

## What did the researchers do and find?

- We used primary and secondary healthcare electronic health records from 837,869 individuals of White, South Asian, and Black African/Caribbean ethnicity in England with early onset of multimorbidity who were registered with an English general practice between 2010 and 2020.

- We found that the early onset of multimorbidity was more common among minority ethnic groups (59% and 56%, in the South Asian and Black populations, respectively) than in the White population (42%) living in the UK.

- South Asians and Black individuals with early onset multimorbidity died earlier than White individuals with early onset multimorbidity (52 and 48 years old in the median, respectively, versus 61 years old).

- South Asian and Black groups were more socioeconomically deprived than White groups, with a consistent deprivation gradient across all multimorbidity clusters. In White groups, the cluster of individuals with the highest rates/odds of the outcomes was more socioeconomically deprived than the cluster with the lowest rates/odds of the outcomes.

## What do these findings mean?

- Our findings emphasise the need to identify, prevent, and manage multimorbidity early in the life course.

- Our work highlights the need to ensure that public health policies are equitable and reach those living in socioeconomic deprivation and minority ethnic groups who are disproportionately and more severely affected by early onset multimorbidity.

## Introduction

The growing prevalence of multimorbidity—the existence of multiple long-term conditions (LTCs) in a single individual [1]—and its burden on individual and population health has recently led to major research investment and health policy initiatives [2,3].

While many previous studies of multimorbidity have focused on older adults [4,5], few have investigated multimorbidity in younger populations [6,7]. Importantly, recent studies have shown an increasing prevalence of multimorbidity in early adulthood [4,8], particularly in ethnic minority and socially deprived populations [8–10]. Relatedly, there is evidence showing that individuals with socioeconomic vulnerability experience poorer health outcomes, such as lower quality healthcare provision, premature death, and higher mortality rates [8–10].

People living with multiple LTCs account for the majority of primary care and hospital utilisation and long-term medication use. Systematic reviews from the United Kingdom [11] and internationally [12] have shown that health service utilisation and costs tend to increase with each additional condition in a single individual. Nonetheless, medical guidelines are centred on the treatment of individual health conditions and often do not account for interactions between conditions that commonly co-occur [13,14]. Likewise, increasing multimorbidity has

been associated with an increased mortality risk, with risk higher still in some ethnic minority groups (Pakistani, Black African, Black Caribbean, and Other Black ethnic groups) compared to the White group [10].

A comprehensive approach to map patterns of multiple LTCs in an ethnically and socioeconomically diverse population with early onset of multimorbidity is crucial to understand the distinct and shared mechanisms that lead to disease accumulation and enable early intervention and reconfigure services to meet the needs of more vulnerable groups.

The majority of studies mapping patterns of multimorbidity focus on diseases as the unit of analysis rather than individuals [15]. However, analysing multimorbidity patterns at an individual level enables a deeper understanding of potentially shared biological and environmental risk factors among specific population groups and understand what similarities they share in terms of sociodemographic profile and what LTCs are the main drivers of increased healthcare service utilisation, long-term prescribing, and mortality.

While multimorbidity is usually defined as the presence of 2 or more LTCs, the majority of multimorbidity research in the UK focuses on a limited set of around 40 highly prevalent LTCs [16–21]. This approach excludes less prevalent or ethnically patterned diseases and is likely to result in a substantial underestimate of the population prevalence of multimorbidity. A recent systematic review also highlighted the variable and poor reporting of multimorbidity and suggested the need for consensus-based, reproducible definitions [22].

In order to address the abovementioned limitations of existing multimorbidity research, the aims of this study were as follows: firstly, to develop and test a consensus-derived, open-access codelist resource for multimorbidity research with an expanded focus on all LTCs that might contribute to multimorbidity, irrespective of prevalence, with a particular ambition that this resource can adequately address ethnic differences in multimorbidity presentations that may be driven by low prevalence but high impact conditions; secondly, to apply a data-driven approach to identify patterns of LTCs in individuals with early onset of multimorbidity in an ethnically and socioeconomically diverse multimorbid population; and thirdly, to assess ethnic differences in the associations between clusters of individuals and 4 clinically meaningful health outcomes: health service utilisation, long-term prescribing, years of life lost (YLL), and mortality.

## Methods

### Study population and data source

We performed a cross-sectional study using the Clinical Practice Research Datalink GOLD (CPRD GOLD), a large representative English electronic health records database [23]. We identified a source population of individuals aged 16 years and over, registered with an English general practice between January 1, 2010, and December 31, 2020, whose data met CPRD's acceptable data quality standards and who had linkage to Hospital Episode Statistics Admitted-Patient Care (HES-APC) data. We also obtained linkage to Office for National Statistics mortality data and area-level deprivation data (Index of Multiple Deprivation (IMD)).

From the CPRD GOLD source population, we selected a study population of individuals who met the following inclusion criteria: (i) had at least 2 out of a list of 204 LTCs; (ii) belonged to one of the following ethnic groups: White, South Asian, or Black African/Caribbean; (iii) had a valid date of second LTC diagnosis in order to calculate age at the onset of multimorbidity; and (iv) had early onset of multimorbidity, defined by having the second LTC recorded between the ages of 16 and 39 years. The selection of our source population is illustrated in Fig 1.

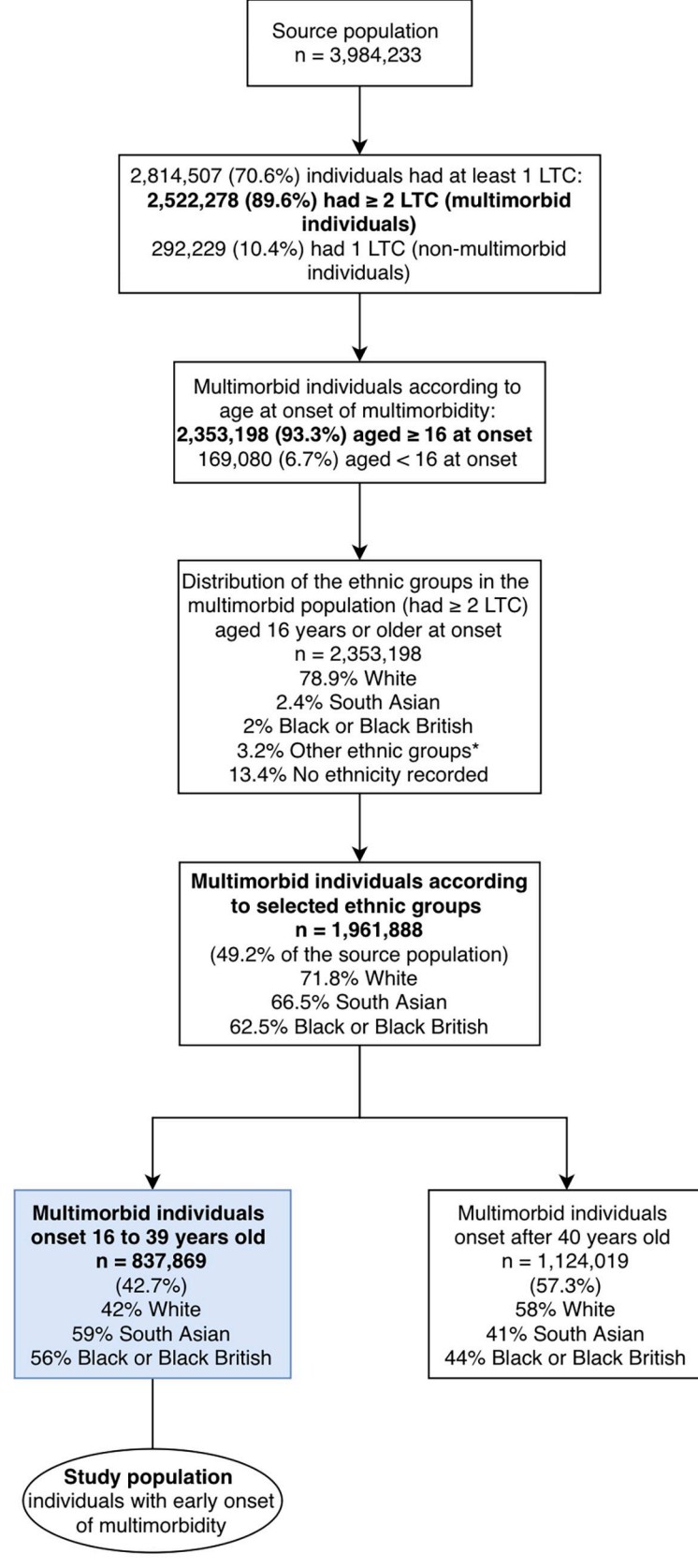

**Fig 1. Flowchart for the selection of our study population.** *Other ethnic groups: Mixed and Chinese and other group.

The use of CPRD data for this study was approved by the Independent Scientific Advisory Committee for the Medicines and Healthcare products Regulatory Agency and the study followed a pre-specific analysis plan (see S1 Protocol and S1 Checklist).

## Identifying the multimorbid population

We used the Academy of Medical Sciences definition of multimorbidity [1], as follows: The coexistence of 2 or more LTCs, each one of which is either (a) a physical noncommunicable disease of long duration, such as cardiovascular disease or cancer; (b) a mental health condition of long duration, such as a mood disorder or dementia; or (c) an infectious disease of long duration, such as HIV or hepatitis C.

Building on existing literature and previous concerns about the lack of reproducibility in multimorbidity research, we undertook a systematic approach to operationalising this definition of multimorbidity for our study. We searched the literature for definitions of multimorbidity and made comparisons between LTCs included in different studies [16,17,24–26]. We searched existing online repositories, publications, and supplementary material for previously built codelists. Where multiple codelists were found, we combined all the relevant codes used by the studies to develop a baseline codelist that underwent extensive clinical review. We conducted a clinical consensus exercise to further refine the set of LTCs to be included in our study (see S1 Text for more details). Detailed information on the methods used to curate the codelists, and the codelists themselves, are available in the MULTIPLY-Initiative online repository [27].

Using all data available from the individual's primary care and secondary care health records, we identified each LTC using the first relevant code ever recorded and calculated the age at diagnosis by subtracting the year of birth from the year of diagnosis. If a condition was never recorded, it was considered absent.

## Stratification groups

To investigate whether the accumulation of LTCs over the life course in people with early onset of multimorbidity differs by ethnicity, we stratified our analysis according to 3 following ethnic groups—White, Black (Black African, Black Caribbean, and Black Other), and South Asian (Indian, Bangladeshi, and Pakistani). Information on self-reported ethnicity was obtained from primary care electronic health records as captured during registration and/or consultation episodes [28].

## Identifying clusters of individuals with early onset multimorbidity

To focus on individuals rather than diseases as the observation in the analysis, we applied latent class analysis to identify groups of people with similar patterns of LTCs accumulation. This approach allowed each LTC to appear in multiple subgroups of individuals, and it is more consistent with clinical experience than other approaches where each LTC could belong to only one cluster at a time. LTCs were identified using all available data in the individual's electronic health record, spanning from as far back as 1920 through to 2020. The latent class analysis is a person-centred mixture modelling approach that identifies latent or unobserved classes (e.g., subpopulations) within a sample based on their patterns of responses to observed variables (e.g., presence/absence of an LTC) given by the posterior membership probabilities, which inform the probability of an individual belonging to a certain subgroup [29]. For each latent class (e.g., subgroup of people with similar characteristics), the average latent class probability was estimated, which indicated the probability of the class model accurately predicting class membership for individuals [30].

For each ethnic group, we tested 2- to 10-class models (where the number of classes represents the number of possible clusters) with a maximum iteration of 1,000 using the

*poLCAParallel* [31] R package and R-4.2.1. We obtained and compared the fit statistics for each model, which, along with clinical judgement, were used to select the optimal number of latent classes for each ethnic group (see S2 Text for more details on the model selection criteria).

## Covariables

Covariables included age in 2010, sex, and socioeconomic deprivation (IMD in quintiles, where the first quintile represents the least deprived areas and the fifth quintile, the most deprived).

## Outcomes

Outcomes included health service utilisation, long-term prescribing, YLL, and mortality. All outcomes were captured between 2010 and 2020. Health service utilisation was defined as the number of primary care consultations (defined by the dates of consultation with any primary care clinician and regardless of the type of consultation), and the number of hospitalisations (defined by the discharge dates related to admitted-patient care) recorded between 2010 and 2020. Long-term prescribing was identified as the counts of unique prescriptions per British National Formulary (BNF) subparagraphs [32], prescribed 3 or more times per year, and it was assessed for the period between 2010 and 2020. Mortality was assessed at the year 5 and 10 after 2010 and was based on the total number of deaths by the end of the respective periods. The YLL were estimated using the remaining average life expectancy after becoming multimorbid and before reaching the average life expectancy at birth of 81 years for the UK population [33].

## Statistical analysis

We described the characteristics of people in our population according to their age in 2010, age at onset of multimorbidity, sex, and socioeconomic deprivation. For each of the clusters of individuals identified from the latent class analysis, we described the distribution of the LTCs per cluster as well as the characteristics of the individual in each cluster according to sociodemographic variables, health service utilisation, YLL, and mortality.

The YLL for each ethnic group and their respective clusters of individuals were estimated using the R library "*lillies*" [34], which allows the estimation of YLL according to a given condition (e.g., groups of individuals with a certain characteristic), and the calculation of confidence intervals using bootstrapping technique.

Generalised linear models adjusted by age in 2010, sex, and deprivation quintile were estimated to investigate which clusters had higher odds of mortality, greater YLL, and higher health services utilisation over a 10-year interval. Odds ratios were estimated using logistic regression models to investigate cluster differences in the odds of mortality by the end of the fifth and 10th years.

Prevalence rate ratios were estimated using negative binomial regression models and zero-inflated Poisson models to account for the overdispersion found in the number of consultations and hospitalisations and to deal with the excess of zeros found in long-term prescribing data. For each outcome of interest, the cluster with the lowest frequency of the outcomes was considered the reference group.

## Results

### Study population

From a total of 3,984,233 people in the CPRD GOLD aged 16 years and over between 2010 and 2020, 2,814,507 individuals (70.6%) had at least one of the 204 LTCs ever recorded in their

electronic health record. We identified our multimorbid population ($n$ = 1,961,888, 69.7%) as those who had developed 2 or more of the 204 LTCs at age 16 years or above, and who belonged to one of the 3 ethnic groups under investigation (Fig 1).

From the total multimorbid population, we identified 837,869 individuals with early onset multimorbidity (16 to 39 years at onset), of whom 777,906 (93%) were White, 33,915 (4%) were South Asian, and 26,048 (3%) were Black African/Caribbean. Early onset multimorbidity was the most common form of multimorbidity among South Asian and Black groups (59%, $n$ = 33,915 and 56%, $n$ = 26,048, respectively) in contrast with the White population (42%, $n$ = 777,906). The median age at multimorbidity onset was 30 years for South Asian, 31 years for Black, and 29 years for White ethnic groups. Women comprised the majority of multimorbid individuals in the South Asian and Black populations (70% and 73%, respectively, compared to 65% in the White population). The early onset multimorbid South Asian and Black populations were mostly from greater socioeconomically deprived areas (28% and 39%, respectively, belonged to the most deprived IMD quintile) compared to the White population where 21% belonged to the least deprived IMD quintile.

The median number of LTCs was higher in the White population (median = 6, IQR 4 to 10), compared to the South Asian (5, 3 to 8) and the Black populations (5, 3 to 8). By the end of the study, 4% ($N$ = 36,027) of the total early onset multimorbid population had died, 4% of White, 2% of South Asian, and 2% of Black African/Caribbean. However, South Asian and Black groups died at a younger age than White groups (median age = 52, 48, and 61 years, respectively) (Table 1).

## Clustering of multimorbid individuals

After evaluating the fit statistics for the latent class models (S2 Text) and upon clinical judgement, we identified 4 clusters of individuals in the White population and 3 clusters of individuals in the South Asian and Black populations. Although we included all 204 LTCs in the latent class models, throughout the paper, we discuss only the 20 most prevalent conditions in each cluster for clarity. The prevalence for all 204 LTCs per cluster and ethnicity can be found in the S1 and S2 Tables, respectively.

Fig 2 shows the distribution of the top 20 most prevalent LTCs across all 3 ethnic groups and the proportion of clusters where each condition appears. Anxiety or phobia, asthma, constipation, depression, dermatitis, enthesopathy, gastro-oesophageal reflux, obesity, and painful conditions were highly prevalent LTCs that occurred in all clusters across the 3 ethnic groups. Eight highly prevalent LTCs appeared exclusively in the White population (alcohol dependence and related disease, chronic obstructive pulmonary disease (COPD), coronary heart disease (CHD), hearing loss, sinusitis, psychoactive substance misuse, urinary incontinence, and venous or lymphatic disease); 4 highly prevalent LTCs appeared exclusively in the South Asian population (polycystic ovarian syndrome, seborrheic dermatitis, thyroid disease, and urticaria); and schizophrenia was highly prevalent in the Black population only (Figs 2 and 3).

## Ethnic differences in healthcare outcomes

During the 10-year period (2010 to 2020), South Asian and White groups had a higher median number of primary healthcare consultations per individual (South Asian 80, IQR: 42 to 143; White 75, 36 to 142), while Black groups had the lowest number of primary care consultations in the same period (71, 36 to 127). Similarly, median number of long-term prescriptions between 2010 and 2020 per individual was higher among South Asian (2, 0 to 5) and White groups (2, 0 to 5) compared to Black groups (1, 0 to 4). However, the median number of hospitalisation episodes over the 10-year period was higher for Black groups (3, 1 to 5) compared to

**Table 1. Sociodemographic characteristics of the population with early onset of multimorbidity according to ethnic group.** CPRD GOLD (2010–2020).

| Ethnic groups | White | South Asian | Black African/Caribbean | Total |
|---|---|---|---|---|
| **Total** | 777,906 (93) | 33,915 (4) | 26,048 (3) | 837,869 (100) |
| **Sex** | | | | |
| Men | 269,668 (35) | 10,263 (30) | 6,910 (27) | 286,841 (34) |
| Women | 508,238 (65) | 23,652 (70) | 19,138 (73) | 551,028 (66) |
| **Age at onset[1]** | | | | |
| Median (Q1–Q3) | 29 (23–34) | 30 (25–34) | 31 (25–35) | 29 (23–34) |
| **Age in 2010** | | | | |
| Median (Q1–Q3) | 37 (28–46) | 32 (27–39) | 33 (27–40) | 36 (28–45) |
| **Age at death[2]** | | | | |
| Median (Q1–Q3) | 61 (49–76) | 52 (43–65) | 48 (41–58) | 61 (49–75) |
| **IMD (quintiles) (%)** | | | | |
| 1Q (least deprived) | 165,180 (21) | 5,343 (16) | 1,614 (6) | 172,137 (21) |
| 2Q | 159,959 (21) | 5,615 (17) | 2,413 (9) | 167,987 (20) |
| 3Q | 159,372 (21) | 6,301 (19) | 4,366 (17) | 170,039 (20) |
| 4Q | 150,768 (19) | 7,173 (21) | 7,418 (29) | 165,359 (20) |
| 5Q (greatest deprived) | 142,020 (18) | 9,466 (28) | 10,213 (39) | 161,699 (19) |
| **Consultation in 10 years** | | | | |
| Median (Q1–Q3) | 75 (36–142) | 80 (42–143) | 71 (36–127) | 75 (36–141) |
| **Hospitalisation in 10 years[3]** | | | | |
| Median (Q1–Q3) | 2 (1–5) | 2 (1–5) | 3 (1–5) | 2 (1–5) |
| **Long-term prescribing in 10 years[4]** | | | | |
| Median (Q1–Q3) | 2 (0–5) | 2 (0–5) | 1 (0–4) | 2 (0–5) |
| **Mortality** | | | | |
| 5-year mortality | 12,906 (2) | 194 (1) | 204 (1) | 13,304 (2) |
| 10-year mortality | 34,922 (4) | 535 (2) | 570 (2) | 36,027 (4) |
| **YLL[5]** | | | | |
| Estimate (95% CI) | 21.1 (21.0–21.2) | 28.0 (27.0–29.2) | 30.8 (29.9–31.9) | 21.4 (21.2–21.6) |
| **Remaining life expectancy** | | | | |
| Estimate (95% CI) | 31.0 (30.8–31.1) | 22.0 (21.2–23.3) | 18.1 (16.9–19) | 31.0 (30.5–31.8) |
| **LTCs** | | | | |
| Median (Q1–Q3) | 6 (4–10) | 5 (3–8) | 5 (3–8) | 6 (4–9) |

[1]Individual's age when the second LTC was recorded.

[2]Individual's age when one of the earliest occur: study end (31 December 2020) or death.

[3]It does not include accident and emergency attendances.

[4]Counts of unique BNF subparagraphs from which an individual had continuous prescribing (3 or more prescriptions that occur in a year).

[5]UK's life expectancy at birth of 81 years old taken as reference.

BNF, British National Formulary; CPRD, Clinical Practice Research Datalink; IMD, Index of Multiple Deprivation; LTC, long-term condition; YLL, years of life lost.

White and South Asian groups (2, 1 to 5). People of Black African/Caribbean ethnicity lost more years of life on average (30.8) than those of White (21.1) and South Asian ethnicity (28.0) after developing early onset multimorbidity and before reaching the average of 81 years old [33].

## Ethnic differences in associations between multimorbidity clusters and outcomes

In each of the 3 ethnic groups, the clusters with the highest rates/odds of the outcomes shared 14 of the 20 most prevalent LTCs and shared their 3 leading conditions—hypertension,

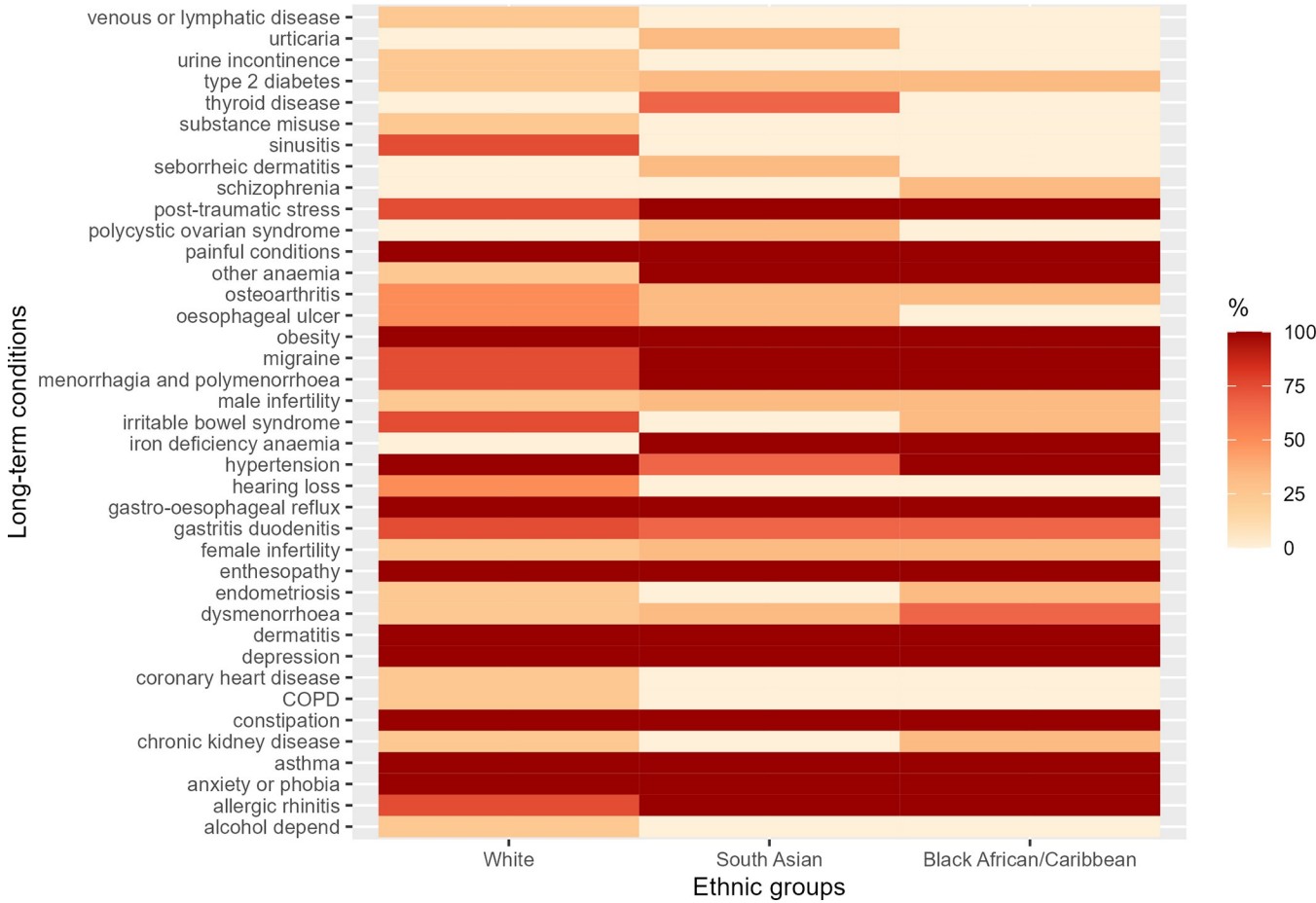

**Fig 2. The top 20 most prevalent overlapping and unique LTCs across all ethnic groups.** The figure shows the overlapping and unique LTC across the 3 ethnic groups and the proportion of clusters they occur within each ethnic group. Maximum number of clusters: White = 4; South Asian = 3; Black = 3.

depression, and painful conditions (Fig 3). However, beyond these similarities, there was variation in the composition of those clusters according to ethnicity: COPD, CHD, hearing loss, and venous or lymphatic disease were prevalent in the White group; allergic rhinitis, iron deficiency anaemia, menorrhagia and polymenorrhoea, migraine, and post-traumatic stress and stress-related disorders were prevalent in South Asian and Black groups; oesophageal ulcer was prevalent conditions in White and South Asian groups, but not in the Black groups; and chronic kidney disease (CKD) was highly prevalent in White and Black groups, but not in the South Asian group (Fig 3). Descriptive sociodemographic characteristics comparisons between the clusters with the highest rates/odds of the outcomes according to ethnic groups can be seen in S1–S3 Figs.

The clusters with the lowest rates/odds of the outcomes were relatively homogeneous across all 3 ethnic groups and were used as the reference group in our regression models. The most common LTCs in those clusters across all ethnicities were female infertility and male infertility.

For the White group, the cluster with the highest rates/odds of outcomes was led by hypertension, depression, and painful conditions (cluster 4), while the cluster with the lowest rates/odds of outcomes was led by female infertility, male infertility, and depression (cluster 1) (Fig 4). Those in

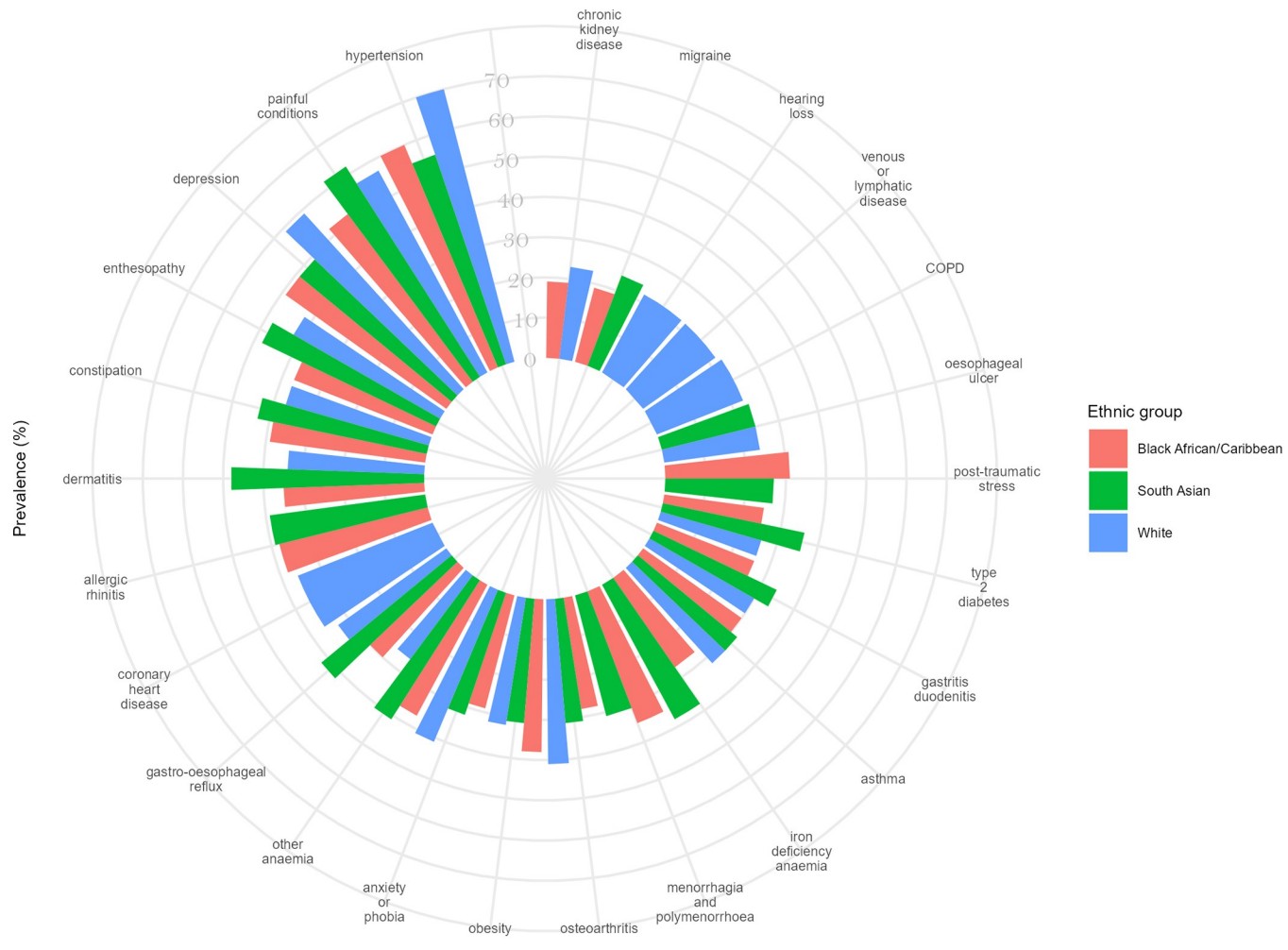

**Fig 3. The top 20 most prevalent LTCs according to clusters within each ethnic group.** Comparison between the clusters with the highest rates/odds for the outcomes in each ethnic group.

cluster 4 had a greater median number of LTCs (*n* = 15, IQR 12 to 20) compared to those in cluster 1 (6, 5 to 9) (Table 2).

Compared to individuals in cluster 1, individuals in cluster 4 had over double the rate of primary care consultation, [PRR = 2.14, 95% CI 2.12 to 2.16]; 3 times the rate of long-term prescribing over 10 years [PRR = 3.15, 95% CI 3.13 to 3.17]); 5 times the rate of hospitalisation [PRR = 5.48, 95% CI 5.45 to 5.51]); and between a 6- to 12-fold higher odds of mortality [OR at the year 5 = 5.93, 95% CI 5.2 to 6.76 and OR at year 10 = 12.03, 95% CI 11.03 to 13.13] (Table 3). They also lost an average of 10.6 years of life after becoming multimorbid and before reaching the UK's life expectancy of 81 years old compared to an average of 0.4 years in cluster 1 (Table 2).

For the South Asian group, the cluster with the highest rates/odds of outcomes was led by painful conditions, hypertension, and depression (cluster 3), while the cluster with the lowest rates/odds of outcomes was led by female infertility, male infertility, and dermatitis (cluster 1) (Fig 5). Individuals in cluster 3 had twice the rate of primary care consultation [PRR = 1.96,

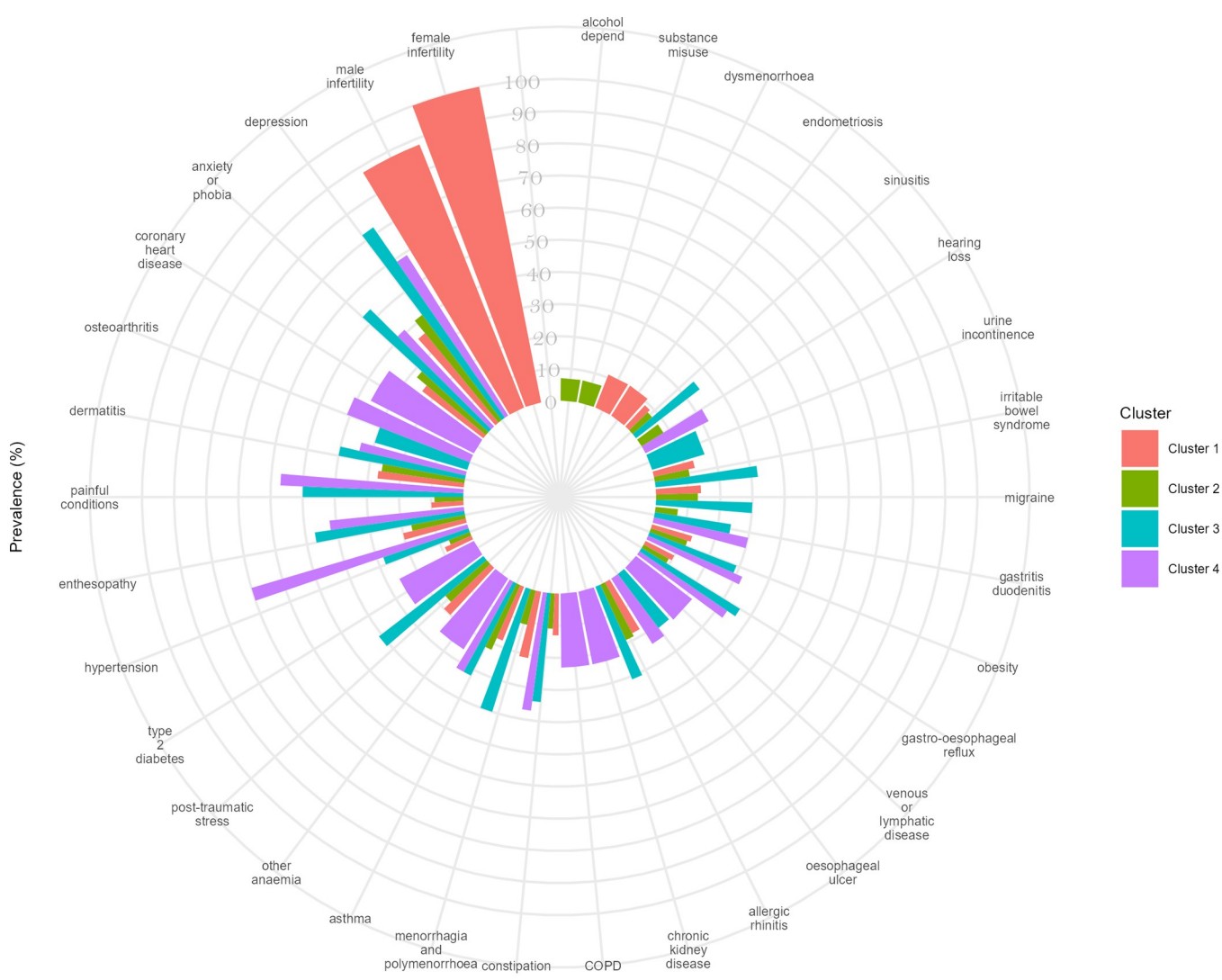

**Fig 4. The top 20 most prevalent LTCs according to clusters in the White population.** The bars display either common or unique LTCs across different clusters within the White population.

95% CI 1.89 to 2.03], twice the rate of long-term prescribing [PRR = 2.57, 95% CI 2.51 to 2.64], 3 times the rate of hospitalisation [PRR = 3.13, 95% CI 3.06 to 3.19], and 8 and 11 times the odds of mortality by the end of the year 5 and 10 [OR = 8.48, 95% CI 3.06 to 23.51; OR = 10.86, 95% CI 5.71 to 20.66, respectively] compared to people in cluster 1 (Table 3). Individuals in cluster 3 lost an average of 18.5 years of life after becoming multimorbid and before reaching 81 years old compared to an average of 0.8 years of life in cluster 1, an order of magnitude greater than the same comparison in Whites (Table 2).

For the Black population, the cluster with the highest rates/odds of outcomes was led by hypertension, painful conditions, and depression (cluster 3), while the cluster with the lowest rates/odds of outcomes was led by female infertility, male infertility, and menorrhagia (cluster 1) (Fig 6). Compared to people in cluster 1, individuals in cluster 3 had higher rates of primary

**Table 2. Characteristics of the clusters of individuals with early onset of multimorbidity from different ethnic groups.**

| | White | | | | South Asian | | | Black African/Caribbean | | |
|---|---|---|---|---|---|---|---|---|---|---|
| | Cluster 1 | Cluster 2 | Cluster 3 | Cluster 4 | Cluster 1 | Cluster 2 | Cluster 3 | Cluster 1 | Cluster 2 | Cluster 3 |
| Individuals in the cluster (%) | 50,202 (6) | 508,742 (65) | 136,604 (18) | 82,358 (11) | 3,963 (12) | 24,245 (71) | 5,707 (17) | 2,754 (11) | 19,553 (75) | 3,741 (14) |
| Average membership (±SD) | 0.97 (±0.08) | 0.96 (±0.1) | 0.86 (±0.16) | 0.91 (±0.14) | 0.99 (±0.05) | 0.98 (±0.07) | 0.93 (±0.12) | 0.99 (±0.06) | 0.98 (±0.07) | 0.92 (±0.13) |
| LTC, median (1Q–3Q) | 6 (5–9) | 4 (3–6) | 11 (10–14) | 15 (12–20) | 6 (4–8) | 4 (3–6) | 14 (11–17) | 6 (4–8) | 4 (3–6) | 12 (10–16) |
| Women (%) | 43,385 (86) | 315,058 (62) | 111,587 (82) | 38,208 (46) | 3,493 (88) | 16,407 (68) | 3,752 (66) | 2,463 (89) | 13,967 (71) | 2,708 (72) |
| Age at onset of multimorbidity[1], median (1Q–3Q) | 29 (24–33) | 28 (22–34) | 30 (24–35) | 33 (28–37) | 29 (25–32) | 29 (25–34) | 32 (27–36) | 31 (27–35) | 30 (25–35) | 33 (28–36) |
| Age in 2010, median (1Q–3Q) | 37 (31–43) | 32 (25–40) | 45 (38–52) | 55 (45–66) | 32 (28–37) | 30 (26–36) | 45 (38–53) | 35 (30–41) | 31 (25–37) | 44 (38–50) |
| Lowest deprivation (%)[2] | 14,099 (28) | 109,032 (21) | 27,188 (20) | 14,861 (18) | 713 (18) | 3,782 (16) | 848 (15) | 191 (7) | 1212 (6) | 211 (6) |
| Greatest deprivation (%)[2] | 6,391 (13) | 91,148 (18) | 26,623 (20) | 17,858 (22) | 1,060 (27) | 6,740 (28) | 1,666 (29) | 1,060 (39) | 7,729 (40) | 1,424 (38) |
| Consultation in 10 years, median (1Q–3Q) | 76 (41–127) | 57 (29–102) | 140 (77–229) | 174 (87–292) | 84 (48–138) | 67 (36–115) | 178 (105–295) | 82 (46–133) | 62 (32–105) | 158 (88–253) |
| Hospitalisation in 10 years, median (1Q–3Q) | 2 (1–4) | 2 (1–4) | 4 (2–7) | 7 (4–13) | 2 (1–4) | 2 (1–4) | 5 (2–10) | 2 (1–5) | 2 (1–5) | 6 (3–11) |
| Long-term prescribing in 10 years[3], median (1Q–3Q) | 1 (0–3) | 1 (0–3) | 5 (2–9) | 9 (5–15) | 1 (0–3) | 1 (0–3) | 9 (5–15) | 1 (0–3) | 1 (0–3) | 7 (3–12) |
| 5-year mortality (%) | 249 (0) | 2,812 (1) | 1,324 (1) | 8,521 (10) | 4 (0) | 51 (0) | 139 (2) | 3 (0) | 73 (0) | 128 (3) |
| 10-year mortality (%) | 564 (1) | 6,658 (1) | 3,655 (3) | 24,045 (29) | 10 (0) | 121 (0) | 404 (7) | 20 (1) | 167 (1) | 383 (10) |
| YLL[4], (95% CI) | 0.4 (0.4–0.5) | 7.5 (7.3–7.6) | 2.6 (2.5–2.7) | 10.6 (10.5–10.7) | 0.8 (0.4–1.3) | 8.6 (7–10) | 18.5 (17.1–19.8) | 1.3 (0.7–1.8) | 11.5 (10.2–13.4) | 18 (16.9–19.2) |

[1]Individual's age when the second LTC was recorded.

[2]Lowest and greatest deprivation: first and fifth quintile of the IMD, respectively.

[3]Counts of unique BNF subparagraphs from which an individual had continuous prescribing (3 or more prescriptions that occur in a year).

[4]UK's life expectancy at birth of 81 years old taken as reference.

BNF, British National Formulary; IMD, Index of Multiple Deprivation; LTC, long-term condition; YLL, years of life lost.

care consultation [PRR = 1.74, 95% CI 1.67 to 1.81] and long-term prescribing [PRR = 2.34, 95% CI 2.27 to 2.41], but the magnitude of difference was less than that seen in the other ethnic groups. However, individuals in cluster 3 had significantly greater rates of hospitalisation [PRR = 5.47, 95% CI 5.35 to 5.6] and the highest odds of mortality by the end of the year 5 [OR = 17.15, 95% CI 5.39 to 54.55] and 10 [OR = 8.32, 95% CI 5.24 to 13.2]. Individuals in cluster 3 lost an average of 18.0 years of life compared to 1.3 years in cluster 1, consistent with the findings from the South Asian population.

We observed different associations between socioeconomic deprivation and outcomes that further varied by ethnicity. While living in socioeconomically deprived areas was associated with lower rates of primary care consultations for South Asians [PRR = 0.92, 95% CI 0.89 to 0.94] compared to their peers from more affluent areas, it was associated with higher rates for Whites [PRR = 1.03, 95% CI 1.02 to 1.04] living in deprived areas compared to their wealthier peers. However, Whites, South Asians, and Black African/Caribbean living in socioeconomically deprived areas had similarly higher rates of hospitalisations [PRR = 1.14, 95% CI 1.13 to 1.14, PRR = 1.19, 95% CI 1.17 to 1.21, and PRR = 1.38, 95%CI 1.35 to 1.42, respectively] and long-term prescribing [PRR = 1.25, 95% CI 1.25 to 1.26, PRR = 1.20, 95% CI 1.18 to 1.23, and PRR = 1.16, 95% CI 1.12 to 1.20, respectively] compared to their peers in less deprived areas

**Table 3. Association between the health service utilisation, mortality, and the different clusters of individuals with early onset of multimorbidity according to ethnic groups.** All models were adjusted by sex, age in 2010, and deprivation. The clusters with the lowest impact on the outcomes were considered as references.

| | Consultation in 10 years PRR (95% CI)[1] | Hospitalisation in 10 years PRR (95% CI)[2] | Long-term prescribing in 10 years PRR (95% CI)[2] | 5-year mortality OR (95% CI)[3] | 10-year mortality OR (95% CI)[3] |
|---|---|---|---|---|---|
| **White** | | | | | |
| **Cluster of individuals** | | | | | |
| Cluster 1 | 1 | 1 | 1 | 1 | 1 |
| Cluster 2 | 0.86 (0.85–0.87)*** | 0.93 (0.92–0.93)*** | 0.93 (0.92–0.94)*** | 1.29 (1.14–1.48)*** | 1.36 (1.24–1.48)*** |
| Cluster 3 | 1.72 (1.7–1.73)*** | 2.01 (1.99–2.02)*** | 2.1 (2.08–2.11)*** | 1.14 (1–1.31)* | 1.49 (1.36–1.63)*** |
| Cluster 4 | 2.14 (2.12–2.16)*** | 5.48 (5.45–5.51)*** | 3.15 (3.13–3.17)*** | 5.93 (5.2–6.76)*** | 12.03 (11.03–13.13)*** |
| **Sex** | | | | | |
| Male (ref.) | 1 | 1 | 1 | 1 | 1 |
| Female | 1.24 (1.24–1.25)*** | 1.24 (1.24–1.25)*** | 1.08 (1.08–1.09)*** | 0.76 (0.73–0.79)*** | 0.83 (0.81–0.85)*** |
| **Age in 2010** | 1.01 (1.01–1.01)*** | 0.98 (0.98–0.98)*** | 1.01 (1.01–1.01)*** | 1.07 (1.06–1.07)*** | 1.06 (1.06–1.06)*** |
| **Socioeconomic deprivation levels[4]** | | | | | |
| Lowest deprivation (ref.) | 1 | 1 | 1 | 1 | 1 |
| Greatest deprivation | 1.03 (1.02–1.04)*** | 1.14 (1.13–1.14)*** | 1.25 (1.25–1.26)*** | 1.98 (1.87–2.1)*** | 2.02 (1.95–2.11)*** |
| **South Asian** | | | | | |
| **Cluster of individuals** | | | | | |
| Cluster 1 | 1 | 1 | 1 | 1 | 1 |
| Cluster 2 | 0.89 (0.87–0.91)*** | 0.95 (0.93–0.97)*** | 0.98 (0.96–1)* | 1.76 (0.63–4.89)* | 1.66 (0.87–3.18)* |
| Cluster 3 | 1.96 (1.89–2.03)*** | 3.13 (3.06–3.19)*** | 2.57 (2.51–2.64)*** | 8.48 (3.06–23.51)*** | 10.86 (5.71–20.66)*** |
| **Sex** | | | | | |
| Male (ref.) | 1 | 1 | 1 | 1 | 1 |
| Female | 1.24 (1.22–1.27)*** | 0.99 (0.97–1.00)** | 1.05 (1.03–1.06)*** | 0.4 (0.3–0.54)*** | 0.41 (0.34–0.49)*** |
| **Age in 2010** | 1.01 (1.01–1.01)*** | 0.99 (0.99–0.99)*** | 1.02 (1.02–1.02)*** | 1.06 (1.04–1.07)*** | 1.06 (1.05–1.07)*** |
| **Socioeconomic deprivation levels[4]** | | | | | |
| Lowest deprivation (ref.) | 1 | 1 | 1 | 1 | 1 |
| Greatest deprivation | 0.92 (0.89–0.94)*** | 1.19 (1.17–1.21)*** | 1.20 (1.18–1.23)*** | 1.07 (0.7–1.64)* | 1.14 (0.87–1.51)* |
| **Black African/Caribbean** | | | | | |
| Cluster 1 | 1 | 1 | 1 | 1 | 1 |
| Cluster 2 | 0.85 (0.82–0.88)*** | 0.93 (0.91–0.95)*** | 0.92 (0.89–0.95)*** | 3.52 (1.1–11.21)*** | 1.2 (0.75–1.92)* |
| Cluster 3 | 1.74 (1.67–1.81)*** | 5.47 (5.35–5.6)*** | 2.34 (2.27–2.41)*** | 17.15 (5.39–54.55)*** | 8.32 (5.24–13.2)*** |
| **Sex** | | | | | |
| Male (ref.) | 1 | 1 | 1 | 1 | 1 |
| Female | 1.27 (1.24–1.30)*** | 1.01 (1.00–1.03)*** | 1.03 (1.01–1.05)*** | 0.51 (0.38–0.67)*** | 0.49 (0.41–0.59)*** |
| **Age in 2010** | 1.01 (1.01–1.02)*** | 0.98 (0.98–0.98)*** | 1.01 (1.01–1.02)*** | 1.05 (1.04–1.06)*** | 1.05 (1.04–1.06)*** |
| **Socioeconomic deprivation levels[4]** | | | | | |
| Lowest deprivation (ref.) | 1 | 1 | 1 | 1 | 1 |
| Greatest deprivation | 0.98 (0.94–1.02)* | 1.38 (1.35–1.42)*** | 1.16 (1.12–1.20)*** | 1.03 (0.58–1.84)* | 1.46 (0.98–2.18)* |

[1] Prevalence Rate Ratios for the coefficients of the negative binomial regression (Overdispersion was found in the outcome variable).

[2] Prevalence Rate Ratios for the coefficients of the zero-inflated Poisson regression (Overdispersion was found in the outcome variable with an excess of zeros).

[3] Odds Ratio for the coefficients of the logistic regression.

* p > 0.05

** p < 0.01

*** p < 0.001

[4] Lowest and greatest deprivation: 1st and 5th quintile of the Index of Multiple Deprivation, respectively.

OR, odds ratio; PRR, prevalence rate ratio.

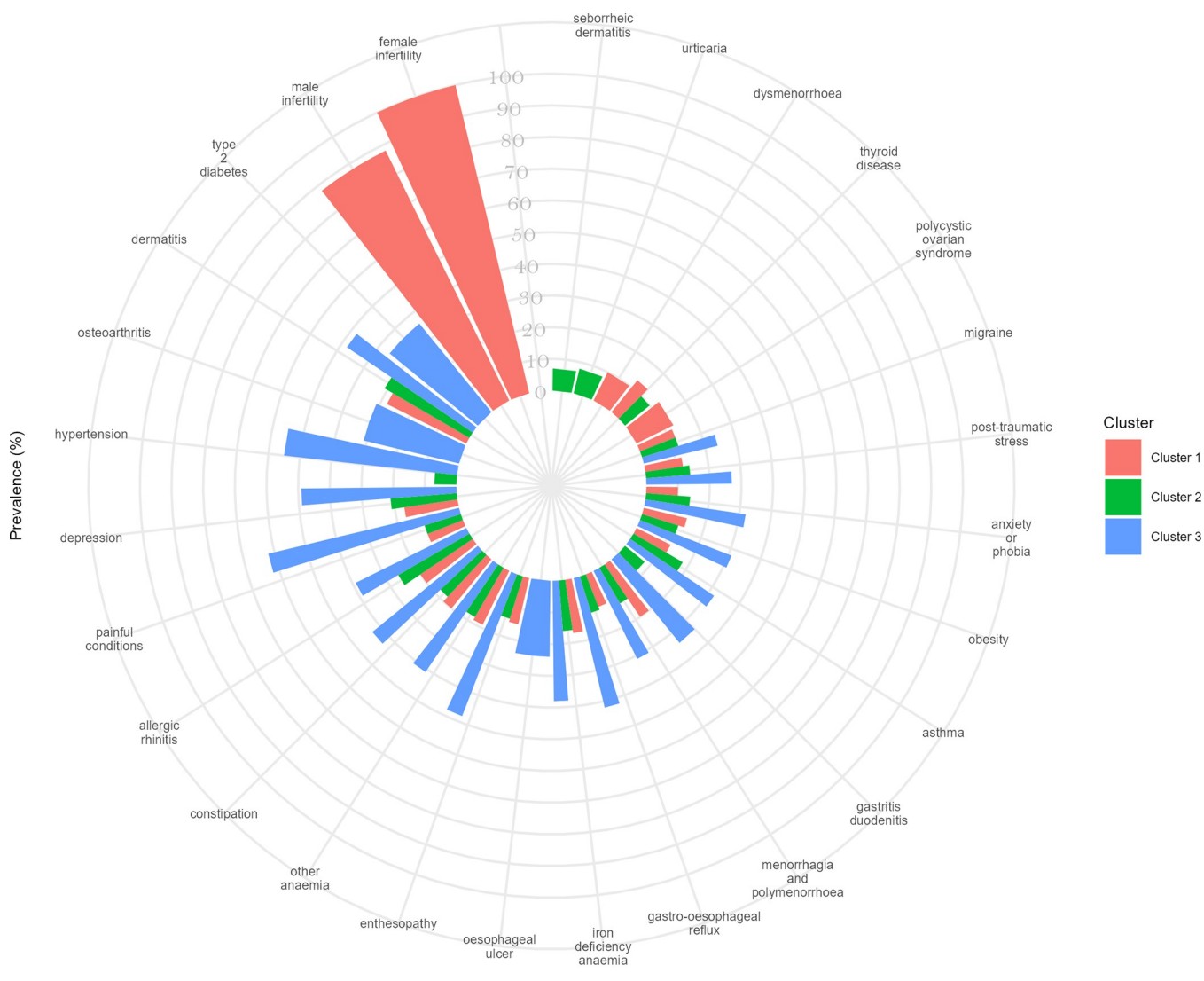

**Fig 5. The top 20 most prevalent LTCs according to clusters in the South Asian population.** The bars display either common or unique LTCs across different clusters within the South Asian population.

(Table 3). These associations assume that the other covariables in the regression model are held constant.

## Discussion

Our findings show that in a large, multimorbid UK population, approximately 40% developed multimorbidity early (aged 16 to 39 years). Black and South Asian populations were more likely to become multimorbid early as compared to Whites. We built on these findings to demonstrate the impact of early onset multimorbidity using measures of healthcare utilisation (primary care consultations, hospitalisation), long-term prescription use, mortality, and YLL. In doing so, we have demonstrated for the first time that South Asian and Black groups with early onset multimorbidity died younger and lost more years of life once they become multimorbid

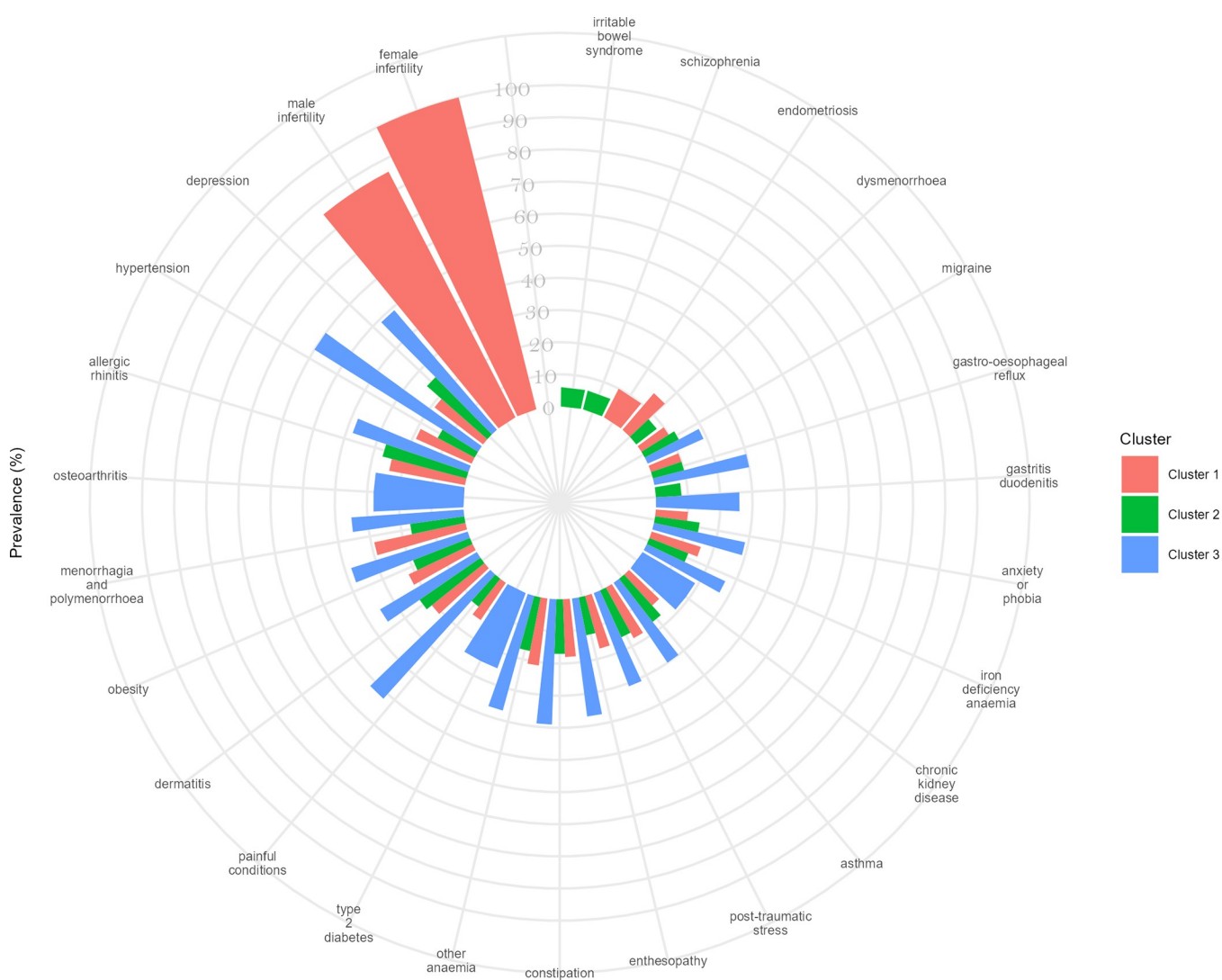

**Fig 6. The top 20 most prevalent LTCs according to clusters in the Black African/Caribbean population.** The bars display either common or unique LTCs across different clusters within the Black African/Caribbean population.

compared to the White group. Although it is well established that multimorbidity increases with age [4,7,35], we show that multimorbidity is highly prevalent in younger populations and disproportionately affects minority ethnic and socially deprived groups, highlighting the need for early interventions to prevent and manage multimorbidity in those populations.

To our knowledge, this is the first study to investigate early onset of multimorbidity and its variation by ethnicity in a large UK population-based sample. By applying a data-driven approach across multiple LTCs, and combining this with measures of healthcare utilisation, mortality, and YLL, we have built new understanding of the burden and impact of early onset multimorbidity at a population scale. Our findings provide important justification to improve the prevention, recognition, and management of multimorbidity in young and diverse populations.

Our observation that the Black population with early onset multimorbidity has the lowest rate of primary care consultations and long-term prescriptions but higher rates of hospitalisation and mortality compared to South Asian and White groups suggests that a lack of routine care could underlie these worse outcomes. However, similar to the Black population, South Asian groups had higher mortality and YLL than the White group yet had the highest rate of primary care consultations across all groups. These findings highlight the importance of understanding the complex relationship between ethnicity, access to and uptake of healthcare in order to improve outcomes from multimorbidity. Previous reports [36–41] have suggested that structural racism may play a role in explaining poorer health outcomes for certain groups within the UK—highlighting less positive experiences of care, insufficient support from local services, poorer treatment outcomes, and a lack of confidence in self-management of multimorbidity among minority ethnic groups compared to White groups [36–41], which is likely to contribute to inequalities in the effective identification and management of multimorbidity, resulting in higher mortality and YLL.

We highlighted important differences in outcomes within ethnic groups associated with deprivation. South Asian people living in areas of high socioeconomic deprivation had lower rates of primary care consultations, but higher rates of hospitalisation and long-term prescribing than their peers living in more affluent areas. Black groups living in areas of high socioeconomic deprivation had higher rates of hospitalisation and long-term prescribing than their peers living in more affluent areas. In contrast, White groups with high socioeconomic deprivation had higher rates of consultation, hospitalisation, and long-term prescribing than their peers from affluent areas. Previous studies show that people with multimorbidity living in more deprived areas may receive poorer quality healthcare represented by shorter consultation times, poorer patient-centeredness, and lower perceived GP empathy compared to those living in more affluent areas [36,42]. These findings demonstrate the intersecting influences of ethnicity and socioeconomic deprivation that require action from clinical and public health systems to tackle upstream determinants of health that contribute to these stark inequalities and poor outcomes from multimorbidity.

Mortality for Black individuals in the cluster led by hypertension, painful conditions, and depression (cluster 3) was greater by the end of year 5 than by the end of year 10. This may suggest a survivor effect in which individuals with more severe health conditions do not reach older ages. Differences in socioeconomic deprivation did not explain the mortality differential between clusters of South Asian and Black individuals. This may be due to the similar distribution of socioeconomic deprivation across all clusters in those ethnic groups, which was different from the White group, where those in the cluster with highest rates/odds of the outcomes had greater levels of socioeconomic deprivation than those in the cluster with lowest rates/odds of the outcomes.

Our comprehensive inclusion of 204 LTCs and the application of data-driven approaches allowed us to generate unique insights into the early onset of multimorbidity, and the contribution of conditions with varied prevalence, e.g., by ethnicity, which are excluded from most other multimorbidity studies (e.g., sickle cell disease, chronic viral hepatitis, polycystic ovarian syndrome, thalassemia). By linking clusters of LTCs to clinically important outcomes, we were able to identify clusters of multimorbid individuals with the highest frequency of primary and secondary care consultations, long-term prescribing, greater YLL, and greater mortality. These clusters in each ethnic group showed some similarities: They largely comprised older people living in areas of high socioeconomic deprivation. Common to all clusters with the highest rates/odds of the outcomes were a range of high prevalence physical and mental health conditions, including hypertension, depression, painful conditions, type 2 diabetes, and anxiety. However, ethnic differences between them were observed. The cluster with the highest rates/

odds of the outcomes in White individuals comprised predominantly men and included conditions not observed in similar clusters in South Asian and Black groups: COPD, CHD, CKD, hearing loss, and venous or lymphatic disease. In contrast, the clusters with the highest rates/ odds of the outcomes in Black and South Asian groups were predominantly comprised of women and included conditions not seen in Whites: allergic rhinitis, iron deficiency anaemia, menorrhagia and polymenorrhoea, migraine, and post-traumatic stress disorder.

Differences in health service use and long-term prescribing associated with multimorbidity may be related to the combination of conditions that lead one to seek healthcare as well as having potentially manageable and resolvable conditions. In addition, individuals with a given LTC are more likely to seek healthcare services, which may result in multiple LTCs being detected over time [12]. Furthermore, the number of LTCs identified may be related to the duration of an individual's data linkage (and, therefore, follow-up time) length of the individuals' follow-ups and age, as individuals with longer follow-up time and older ages have had time and opportunity to have their health-related conditions detected.

Guidelines in the UK that address and manage multimorbidity do not yet include guidance for managing the accumulation of LTCs in individuals with an early onset, nor bring guidance on targeted healthcare for minoritized ethnic groups or socioeconomically deprived groups at high premature risk for poorer health outcomes such as hospitalisation and premature mortality [43]. Public health policies that aim to reduce multimorbidity should be applied in younger populations and, although universal, should increase targeting towards minority groups and the more socioeconomically disadvantaged population.

## Strengths and limitations

The major strength of our study is the large scale of the population studied and the application of data-driven analyses across a robustly defined set of 204 LTCs, building significantly on the existing evidence base. The strength of this approach has enabled us to ensure thatwe represent diseases that may be ethnically patterned and which may contribute differently to the significant burden of early onset multimorbidity. Our approach elucidates early onset clusters of multimorbidity that confer particularly high rates/odds of poor outcomes, and the identification of these provides a rationale for developing improved clinical pathways for the prevention and management of multimorbidity.

Our analysis focused on patients rather than diseases as the unit of analysis allowing for a deeper understanding of patient groups that share patterns of conditions and may provide essential information for the development of clinical guidelines and pathways of care. Although there is no "gold standard" on the selection of multiple LTCs for multimorbidity studies, we have shown that multimorbidity definitions can be operationalised in electronic health records, and our efforts have contributed to enhancing robust reproducible methodology.

There are some limitations of our work. The cross-sectional study design is susceptible to reverse causality, which occurs when the exposure and outcome are measured at the same point in time, and there is no clarity about which event occurred first. In our study, the exposure (clusters of LTCs) was measured at any time up to 2020, and the outcomes were measured between 2010 and 2020. Although around 79% of the exposure (e.g., diagnosis of the LTCs) occurred before 2010, we cannot be certain about the temporality of the events—whether all clusters were developed before or after the episodes of consultations, hospitalisations, and long-term prescribing. Despite the uncertainty of those events' temporality, it is likely that those events are interconnected considering that a LTC diagnosis might be ascertained over a consultation or hospitalisation and the use of long-term prescribing may occur to treat a given

health condition. Residual confounding may be present as there are likely to be other unmeasured factors underlying the association between multimorbidity and poorer outcomes that we were unable to study in our analysis (e.g., educational level and aetiological factors). Another limitation is that 13.4% of the multimorbid population had no ethnicity recorded and may be inherently different (likely to have nonrandom missing data) and have poorer health than those with an ethnic group recorded [44]. Besides, other ethnic groups were not included in the analysis, for example, Mixed and Chinese or other group. The reason was the small population size within those ethnic group that fulfilled our study eligibility criteria, resulting in inadequate power to derive robust clusters of multimorbidity and examine associations with health outcomes and mortality.

There are some limitations intrinsic to the use of electronic health records. The health conditions selected might be subject to misclassification due to unrecorded, miscoded, and undiagnosed diseases. The age of onset might not reflect the actual age at which a given condition was diagnosed, but the date when it was entered in the patient's medical record. Additionally, the number of LTCs identified is related to the use of the healthcare service, as well as the duration of data availability within their electronic health records. Individuals with data recorded for longer periods have had time and opportunity to have multimorbidity detected. Finally, there may be a time-related bias in the LTCs coding given that we included any diagnosis code ever recorded and therefore, there are fewer codes being recorded particularly prior to late 1990 when 96% of the general practices were using computerised record systems [45].

## Conclusions

Early onset multimorbidity is the most common form of multimorbidity among minority ethnic populations in the UK. Across ethnicities, the clusters with highest rates/odds of the outcomes were common in more socioeconomically deprived individuals and contained several common long-term physical and mental health conditions including hypertension and depression. However, they also demonstrated variability between ethnic groups by sex and conditions. It is likely that the worse outcomes from early onset multimorbidity in minority ethnic and socioeconomically deprived groups may, in part, be due to receiving poorer routine healthcare. These findings emphasise the need to identify, prevent, and manage multimorbidity early in the life course. Most health systems remain focused on single disease management, and our findings add further weight to calls to restructure healthcare provision to do so. Our work provides additional insights into the need to ensure these healthcare improvements are equitable and reach those from socioeconomically deprived and diverse groups who are disproportionately and more severely affected by multimorbidity.

## Supporting information

**S1 Protocol. Study protocol for research using the Clinical Practice Research Datalink (CPRD).**
(PDF)

**S1 Checklist. STROBE Statement. Checklist of items that should be included in reports of cross-sectional studies.**
(PDF)

**S1 Text. Selection of the LTCs.**
(DOCX)

**S2 Text. Fit statistics for the model selection.**
(DOCX)

**S1 Table. Prevalence of the 204 LTCs according to clusters within each ethnic group.**
(DOCX)

**S2 Table. Prevalence of the 204 LTCs according to ethnic groups.**
(DOCX)

**S1 Fig. Characteristics of the clusters with the highest rates/odds of the outcomes according to ethnic group.** Set of boxplots showing the interquartile range, minimum and maximum values, and outliers for age at onset, age in 2010, consultation in 10 years, hospitalisation in 10 years, continuous therapy in 10 years, and LTC.
(PNG)

**S2 Fig. Characteristics of the clusters with the highest rates/odds of the outcomes according to ethnic group.** Boxplots showing the interquartile range, minimum and maximum values, and outliers for age at death and YLL.
(PNG)

**S3 Fig. Characteristics of the clusters with the highest rates/odds of the outcomes according to ethnic group.** Histogram showing the proportion of people who died in the year 10 and 5, the proportion of females and people at the greatest and lowest socioeconomic deprivation levels.
(PNG)

## Acknowledgments

The authors would like to thank Sherman Lo, research software engineer in the ITS Research, Queen Mary University of London, for assistance in expanding and improving the poLCA R library by creating the poLCAParallel R package and enabling a much faster run of our codes.

## Disclaimer

This study is based in part on data from the Clinical Practice Research Datalink obtained under licence from the UK Medicines and Healthcare products Regulatory Agency. The data are provided by patients and collected by the NHS as part of their care and support. The interpretation and conclusions contained in this study are those of the authors alone. The ONS is the provider of the ONS Data contained within the Dataset. HES and ONS data were reused with the permission of The Health & Social Care Information Centre. The OPCS Classification of Interventions and Procedures, codes, terms, and text is Crown copyright (2016) published by Health and Social Care Information Centre, also known as NHS Digital and licenced under the Open Government Licence available at www.nationalarchives.gov.uk/doc/open-government-licence/open-governmentlicence.

## Author Contributions

**Conceptualization:** Fabiola Eto, Miriam Samuel, Sally Hull, Sarah Finer, Rohini Mathur.

**Data curation:** Fabiola Eto, Miriam Samuel, Rafael Henkin, Meera Mahesh, Tahania Ahmad, Alisha Angdembe, R. Hamish McAllister-Williams, Michael R. Barnes, Sarah Finer, Rohini Mathur.

**Formal analysis:** Fabiola Eto, Rafael Henkin, Alisha Angdembe, Sarah Finer.

**Funding acquisition:** Sally Hull, Sarah Finer, Rohini Mathur.

**Investigation:** Sarah Finer, Rohini Mathur.

**Methodology:** Fabiola Eto, Miriam Samuel, Rafael Henkin, Paolo Missier, Nick J. Reynolds, Michael R. Barnes, Sally Hull, Sarah Finer, Rohini Mathur.

**Project administration:** Sarah Finer.

**Resources:** Sarah Finer.

**Supervision:** Sarah Finer, Rohini Mathur.

**Validation:** Fabiola Eto, Sarah Finer.

**Visualization:** Fabiola Eto.

**Writing – original draft:** Fabiola Eto, Sarah Finer, Rohini Mathur.

**Writing – review & editing:** Fabiola Eto, Miriam Samuel, Rafael Henkin, Meera Mahesh, Tahania Ahmad, Alisha Angdembe, R. Hamish McAllister-Williams, Paolo Missier, Nick J. Reynolds, Michael R. Barnes, Sally Hull, Sarah Finer, Rohini Mathur.

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
