## [Editor Report · Decision Letter 0]

15 Mar 2023

Dear Dr Eto, 

Thank you for submitting your manuscript entitled "Ethnic differences in early onset multimorbidity and associations with health service use, long-term prescribing, years of life lost, and mortality an observational study using person-level clustering in the UK Clinical Practice Research Datalink" for consideration by PLOS Medicine.

Your manuscript has now been evaluated by the PLOS Medicine editorial staff and I am writing to let you know that we would like to send your submission out for external assessment.

However, before we can send your manuscript for assessment, we need you to complete your submission by providing the metadata that are required for full assessment. To this end, please login to Editorial Manager where you will find the paper in the 'Submissions Needing Revisions' folder on your homepage. Please click 'Revise Submission' from the Action Links and complete all additional questions in the submission questionnaire.

Please re-submit your manuscript within two working days, i.e. by Mar 17 2023 11:59PM.

Once your full submission is complete, your paper will undergo a series of checks in preparation for external assessment.

Kind regards,

Richard Turner PhD

Consulting Editor, PLOS Medicine

plosmedicine@plos.org

---

## [Decision Letter · Decision Letter 1]

10 Jul 2023

Dear Dr. Eto,

Thank you very much for submitting your manuscript "Ethnic differences in early onset multimorbidity and associations with health service use, long-term prescribing, years of life lost, and mortality: an observational study using person-level clustering in the UK Clinical Practice Research Datalink" (PMEDICINE-D-23-00573R1) for consideration at PLOS Medicine. 

Your paper was discussed with an academic editor with relevant expertise and sent to independent reviewers, including a statistical reviewer. The reviews are appended at the bottom of this email and any accompanying reviewer attachments can be seen via the link below:

[LINK]

In light of these reviews, we will not be able to accept the manuscript for publication in the journal in its current form, but we would like to invite you to submit a revised version that addresses the reviewers' and editors' comments fully. You will appreciate that we cannot make a decision about publication until we have seen the revised manuscript and your response, and we expect to seek re-review by one or more of the reviewers. 

We hope to receive your revised manuscript by Jul 31 2023 11:59PM. Please email us (plosmedicine@plos.org) if you have any questions or concerns.

Please let me know if you have any questions, and we look forward to receiving your revised manuscript. 

Sincerely,

Richard Turner PhD

Consulting editor, PLOS Medicine

plosmedicine@plos.org

In the abstract (Methods and findings subsection), please quote aggregate demographic and ethnic details for study participants. 

Where feasible, please quote numbers of participants along with the corresponding percentages in the abstract. 

Please add a new final sentence to the 'Methods and findings' subsection of your abstract, which should begin "Study limitations include ..." or similar and should quote 2-3 of the study's main limitations. 

After the abstract, please add a new and accessible, tripartite 'Author summary' section in non-identical prose. You may find it helpful to consult one or two recently published research papers published in PLOS Medicine to get an impression of the preferred style. 

In the Methods section (main text), please state whether or not the study had a protocol or prespecified analysis plan, and if so attach the relevant document(s) as supplementary file(s), referred to in the text, 

We suggest moving the participant flowchart to the main body of the paper.

Noting referee 1's comments on p values, we ask that these remain in the relevant table in accord with PLOS Medicine policy. 

Please restructure the early part of the Discussion (main text): the first paragraph should consist predominantly of a summary of the main findings, and other elements can appear in subsequent paragraphs. 

Noting reference 1 and others, please list an individual or group author name or names first, as for the scholarly references. 

Noting references 9 & 10, for example, please use the journal name abbreviations "PLoS ONE." and "PLoS Med.".

Please add a completed checklist for the most appropriate reporting guideline, e.g., STROBE, labelled "S1_STROBE_Checklist" or similar and referred to in the Methods section (main text). 

In the checklist please refer to individual items by section (e.g., 'Methods') and paragraph number, but by line or page numbers as these generally change in the event of publication. 

Comments from the reviewers:

*** Reviewer #1: 

Thanks for the opportunity to review your manuscript. My role is as a statistical reviewer, so my review concentrates on the study design, data, and analysis that are presented. I have put general questions first, followed by queries relevant to a specific section of the manuscript (with a page/paragraph reference, going from p1 = abstract page).

This study uses data from the UK CPRD (which covers England). Patients with early multimorbidity (<39 years with 2 conditions) and appropriate data (dates of diagnoses recorded and belonging to white/south Asian/black or black British ethnic background). Multimorbidity could include any condition that was 'long term', including non-communicable, infectious, and mental health conditions. The list of conditions was based on a consensus approach with clinicians. A latent class analysis was used to estimate groups of related comorbidities - this was done separately for each ethnic background. 4 latent classes were found for those with a White background, and 3 for South Asian and Black/Black British. Several outcomes were used, these were health service utilisation (primary care consultations, and number of hospitalisations), long-term prescribing (counts of unique prescription, recorded 3 or more times in a year), YLL and mortality. 

I like the approach to classifying multimorbidity, and the rigour which you applied to the process of compiling the lists of conditions. Most of the applications of statistical methodology is appropriate, I think most of the queries below could be dealt with by some minor modifications in a revision or further information provided.

I was just after some clarity with regards to the lookback period for multimorbidity and cohort inception. It looks as though cohort inception was at the beginning of 2010 - was all available data used for the lookback period to ascertain multimorbidity? What potential is there from the CPRD to get misclassified condition information if there are different periods of lookback available for participants with different ages at cohort inception? 

I wasn't clear whether condition information was from information collected during hospitalisation episodes or diagnoses recorded on the EHR by GP/specialists. Were all EHR records used to get these diagnoses? 

One limitation is the interpretation of the coefficients from the multivariable adjusted analysis. The focus of this manuscript seems to be on differences between the clusters in the outcomes, but depending on the causal relationship between the other covariates the same interpretation can't be made of the coefficients from the other covariates, i.e. Table 2 fallacy (https://doi.org/10.1093/aje/kws412). 

Most of the manuscript is written without focusing on p-values for inference, I think this is a good approach given the sample size and would suggest the p-value indicators be taken from Table 3.

Page 5, Paragraph 4. I would typically test a '1 class model', and it is possible that this is the best fitting number of classes (i.e. there are no latent classes). Can I confirm that the 1 class model was not a better fit than any of the models with >=2 classes? 

What criteria was used to decide on the upper limit to the number of classes? 

Was the criteria for selecting number of classes (at a point where there is relatively small changes in BIC etc. and an improvement in entropy relative to adjacent number of classes) decided a priori or was this decided after seeing the fit indices? 

Page 6, Paragraph 1. For reference class, could this be better described as the lowest rate of outcome? 

What criteria was used to decide the neg-bin and ZIP models for the different outcomes? Did the residuals from these models indicate an appropriate distribution (this might be hard to see with your sample size!). 

Page 7, Paragraph 2. Is there an overall descriptor that could be used to distinguish within high/low clusters? This might not be possible but if there is, it does make the analysis using the LC class as an exposure easier to follow. 

Figure 1. It looks like the order of conditions on the y-axis is just in alphabetical order - could a meaningful order such as overall prevalence be used to order this axis to make it more meaningful? 

Table 3+4. Is Cluster 1 a 'medium risk' cluster? 

*** Reviewer #2: 

Thank you for the opportunity to review this work. The authors of this study presenting an important analysis of clusters of multimorbidity among younger adults, how these vary by ethnicity, and their association with a multitude of outcomes. It was an interesting piece to review and I commend the authors for their important work. For context, to some of these points I am a reader in the United States.

I have a few minor comments they may wish to consider:

1. Overall, this piece is incredibly lengthy. I would challenge the authors to perhaps find areas in which they could be more concise. Importantly, for example, the conclusion should be shortened and should not contain hypotheses. Instead it should focus on summarizing the work at hand. 

2. Acronyms: Throughout the manuscript there are several acronyms that made it hard to follow. "LTC" has varied meanings and in the US they are referred to as "chronic conditions." While this may not be the authors preferred term, I would encourage not using the LTC acronym given the use of LCA. Similarly, HRC and LRC seem like unnecessary acronyms that made it hard to follow. Finally, please note "IMD" as an acronym in the "Covariables" section.

3. In the United States, the groups would be referred to as "race" and not "ethnicity." While I understand this social construct varies across countries, I just want to make sure this is consistent with the terminology expected by an English audience. For example, is the variable labelled as ethnicity? Similarly, terms such as "Blacks" should not be used.

4. It is unclear, in the introduction, methods, and discussion, what is meant by the fact that most multimorbidity studies focus on diseases. Please elaborate what this means conceptually and methodologically.

5. I am not convinced that the statement "is more likely to reveal underlying disease relationships within clusters" is entirely accurate. I'm not sure that a data driven approach, including LCA, will truly do this from a conceptual perspective.

6. Could the authors comment on if there were ethnic groups that were not included? Other than those with missing.

7. I have some trepidation with the use of YLL as an outcome. I think, while it is certainly interesting, there are perhaps some methodological limitations in this estimation. Given that bootstrapping is mentioned, is this calculated based on the sample and the distribution of individuals who were known to have died? Which is just 2-4%. If the authors wish to keep this in I think a more clear walk through of this estimation and discussion of the limitations is needed.

8. While we consider long-term prescribing perhaps a negative outcome generally, perhaps in this context it is a positive outcome? Given that the sample was limited to those with early onset multimorbidity is prescribing a sign of appropriate care? Further, given that the types of medications were not examined it's hard to say if this is truly a bad measure. Perhaps it is a marker of disease severity but is similarly a marker of appropriate treatment. I would challenge the authors to think a bit more about this in the context of this work.

9. Figure 1: Given that, based on my understand, the % being represented is out of 4 (White) or 3, I'm not sure the continuous scale makes the most sense. Would some type of discrete scale be more applicable here? While I understand what the figure is conveying, seeing as it's not truly a continuous measure the shading of the colors is a bit more challenging to follow. 

*** Reviewer #3: 

Dear editor,

Thank you for allowing me to read this very interesting article regarding ethnic differences in early onset multimorbidity and associations with health service use, long-term prescribing, years of life lost, and mortality.

I believe that the manuscript is well written and that the research was presented in a way that is easy to follow and understand. I think the statistics were adequately explained and their meaning was easy to understand.

The authors have concluded that the worse outcomes from early onset multimorbidity in Black and socio-economically deprived groups may, in part, be due to receiving poorer routine healthcare. This conclusion is right on the money and has the possibility to influence healthcare services in achieving equity.

Congratulations to the authors. Hope to see more on this research line in the future. 

*** Reviewer #4: 

Thank you for the opportunity to read this interesting and relevant manuscript. This work provides important insight into socioeconomically deprived and diverse populations, particularly focusing on multimorbidity experienced early in the life source, where there may be more potential for intervention, as well as hypothesis generation into mechanisms of condition accrual. I have only a few comments.

Introduction. The second last paragraph may not be necessary in this section and extra detail could be consolidated with additional methods description or moved to supplementary.

Methods. Could some of the detail be made more concise in the main text and the full description be moved to supplementary, for example paragraph two, and details about the process used to defined multimorbidity. Which ethnic groups were excluded from the study and how many people did this involve - this is in supplementary but might be useful to document in methods (with %). Could any of the conditions be double counted, could this be a limitation when looking at multimorbidity but could be a strength when looking at condition accrual. Access to the code lists publicly would be very helpful for other researchers.

Results. Some of the results section text could be in methods/supplementary, for example paragraph 4 "After evaluating first statistics for the latent class models.." where methods are described.

***

[LINK]

---

## [Decision Letter · Decision Letter 2]

10 Sep 2023

Dear Dr. Eto,

Thank you very much for re-submitting your manuscript "Ethnic differences in early onset multimorbidity and associations with health service use, long-term prescribing, years of life lost, and mortality: a cross-sectional study using clustering in the UK Clinical Practice Research Datalink" (PMEDICINE-D-23-00573R2) for consideration at PLOS Medicine.

I have discussed the paper with our academic editor and it was also seen again by two reviewers. I am pleased to tell you that, provided the remaining editorial and production issues are fully dealt with, we expect to be able to accept the paper for publication in the journal.

[LINK]

Please let me know if you have any questions, and we look forward to receiving the revised manuscript.   

Sincerely,

Richard Turner PhD

Consulting Editor, PLOS Medicine

plosmedicine@plos.org

Requests from Editors:

At line 43, for example, please use square brackets ("... conditions [LTCs] ...") within brackets. 

At line 53, please substitute "identified", or similar, for "unveiled".

At line 59, please revisit "the White population was predominantly male ...". Perhaps some information about the group referred to is needed (otherwise this statement would seem inconsistent with information in table 1). 

At line 64, please adapt the punctuation to: "... respectively); however, the ...".

At line 123, please make that "the White group".

At line 140, please substitute a colon for the semicolon. 

At line 162 and any other instances, please make that "... 39 years".

At line 211, "age in 2010"?

At line 255, we suggest quoting the number for the White population with early-onset multimorbidity alongside the percentage. 

At line 310, please substitute a colon for the semicolon. 

At line 354, please make that "people in cluster 1".

At line 359, "lost an average"?

At line 376, we suggest substituting "... as compared to Whites".

At line 415, we suggest making that "year 5" and "year 10".

At line 443, there may be some repetition in "older ages and older": please check.

At line 465, "reverse causality"?

At line 489, please make that "Finally ...".

At line 493, "minority ethnic populations"?

In table 1, second column, row 7, adapt to "165,180"?

Please remove the information on data sharing from the end of the main text. In the event of publication, this will appear in the article metadata, via entries in the submission form. 

Noting reference 2, please remove all iterations of "[Internet]" from the reference list.

Comments from Reviewers:

*** Reviewer #1: 

Thanks for the revised manuscript and responses to my queries. The updates to the manuscript have resolved my original questions - this is an interesting study I enjoyed reviewing. I appreciate the information on the parallel version of poLCA, I have been through the pain of LCA with a large dataset before. 

*** Reviewer #2: 

I thank the authors for their thoughtful replies to my comments. I have nothing further.

***

[LINK]

---

## [Editor Report · Decision Letter 3]

17 Sep 2023

Dear Dr Eto, 

On behalf of my colleagues and the Academic Editor, Dr Basu, I am pleased to inform you that we have agreed to publish your manuscript "Ethnic differences in early onset multimorbidity and associations with health service use, long-term prescribing, years of life lost, and mortality: a cross-sectional study using clustering in the UK Clinical Practice Research Datalink" (PMEDICINE-D-23-00573R3) in PLOS Medicine.

Prior to final acceptance, please address the following points:

In the final sentence of the 'Background' subsection of the abstract, please adapt the wording so as to state the study aim rather than the findings (e.g., "... diverse population, aiming to identify associations between ...").

At line 71 in the abstract, and in the main text, you use the term "Chinese ... ethnic group[s]". Please consider adapting this to "Southeast Asian ethnicity" or "Han ethnicity" as appropriate. 

PRESS

Sincerely, 

Richard Turner PhD

Consulting Editor, PLOS Medicine

plosmedicine@plos.org